# Structural Insight into a Yeast Maltase—The *Ba*AG2 from *Blastobotrys adeninivorans* with Transglycosylating Activity

**DOI:** 10.3390/jof7100816

**Published:** 2021-09-29

**Authors:** Karin Ernits, Christian Kjeldsen, Karina Persson, Eliis Grigor, Tiina Alamäe, Triinu Visnapuu

**Affiliations:** 1Department of Chemistry, Umeå University, 90187 Umeå, Sweden; karin.ernits@umu.se (K.E.); karina.persson@umu.se (K.P.); 2Department of Chemistry, Technical University of Denmark, DK-2800 Kgs. Lyngby, Denmark; chkje@kemi.dtu.dk; 3Institute of Molecular and Cell Biology, University of Tartu, 51010 Tartu, Estonia; eliis.grigor25@gmail.com (E.G.); tiina@alamae.eu (T.A.)

**Keywords:** α-glucosidase, glycoside hydrolase, isomalto-oligosaccharides, acarbose, crystal structure, molecular replacement, nuclear magnetic resonance, erlose, isomelezitose, trehalulose

## Abstract

An early-diverged yeast, *Blastobotrys* (*Arxula*) *adeninivorans* (*Ba*), has biotechnological potential due to nutritional versatility, temperature tolerance, and production of technologically applicable enzymes. We have biochemically characterized from the *Ba* type strain (CBS 8244) the GH13-family maltase *Ba*AG2 with efficient transglycosylation activity on maltose. In the current study, transglycosylation of sucrose was studied in detail. The chemical entities of sucrose-derived oligosaccharides were determined using nuclear magnetic resonance. Several potentially prebiotic oligosaccharides with α-1,1, α-1,3, α-1,4, and α-1,6 linkages were disclosed among the products. Trisaccharides isomelezitose, erlose, and theanderose, and disaccharides maltulose and trehalulose were dominant transglycosylation products. To date no structure for yeast maltase has been determined. Structures of the *Ba*AG2 with acarbose and glucose in the active center were solved at 2.12 and 2.13 Å resolution, respectively. *Ba*AG2 exhibited a catalytic domain with a (β/α)_8_-barrel fold and Asp216, Glu274, and Asp348 as the catalytic triad. The fairly wide active site cleft contained water channels mediating substrate hydrolysis. Next to the substrate-binding pocket an enlarged space for potential binding of transglycosylation acceptors was identified. The involvement of a Glu (Glu309) at subsite +2 and an Arg (Arg233) at subsite +3 in substrate binding was shown for the first time for α-glucosidases.

## 1. Introduction

*Blastobotrys adeninivorans* (syn. *Arxula adeninivorans*) is a non-conventional yeast species of Saccharomycotina subphylum which diverged in the evolution of fungi long before *Saccharomyces* sp. ([1] and references therein). *B. adeninivorans* is considered biotechnologically valuable as it accumulates lipids [2,3], is thermo- and osmotolerant and metabolically versatile—for growth it uses numerous carbon and nitrogen sources atypical for yeasts [4,5]. Being dimorphic, *B. adeninivorans* turns from budding cells to a filamentous form when cultivated above 42 °C, which also promotes enzyme secretion [6]. Several biotechnologically relevant enzymes of *Blastobotrys* sp., such as cutinases [7], tannase [8], glucoamylase [9], invertase [10], maltase [1], and purine-degrading enzymes [11], have been isolated and characterized. Thus far, only the structure of the gallic acid decarboxylase AGDC1 from *Blastobotrys* sp. has been determined [12]. Based on the AGDC1 structure, the authors suggested a novel decarboxylation mechanism that combines acid-base catalysis and transition-state stabilization [12].

In 2020, we characterized the maltase *Ba*AG2 from the type strain of *B. adeninivorans* [1]. This enzyme efficiently hydrolyzed substrates common for maltases (for example maltose and sucrose, malto-oligosaccharides), while atypically for yeast maltases it had a low but clearly recordable exo-hydrolytic activity on amylose, amylopectin, and glycogen. *Ba*AG2 had a considerable transglycosylating ability on maltose to produce potentially prebiotic di- and trisaccharides: panose, maltotriose, and isomaltose [1]. In accordance with the large evolutionary distance between *B. adeninivorans* and other yeasts for which maltases have been characterized, protein sequence identity of *Ba*AG2 with maltases of other yeasts was only moderate, and was shown to be the highest (51%) with the maltase MalT of a filamentous fungus *Aspergillus oryzae* [1,13].

The biotechnological importance of several fungal α-glucosidases (maltases, maltases-isomaltases) lies in their transglycosylation ability. In the transglycosylation reaction otherwise rare oligosaccharides, such as erlose, melezitose, theanderose, isomaltose, maltulose, and turanose, can be produced [1,14,15,16,17]. The same di- and trisaccharides were found in samples of honey [18,19,20]. These sugars have potential applications as low-calorie sweeteners, functional food components and prebiotics. There are some reports on α-glucosidases of yeast and filamentous fungi that synthesize (iso)malto-oligosaccharides from maltose [1,14,15]. It has been shown that transglycosylation of sucrose by yeast maltases results in a non-reducing trisaccharide—isomelezitose [16,17,21]. Recently characterized yeast α-glucosidases from *Metschnikowia* sp. synthesized several oligosaccharides from sucrose: isomelezitose, erlose, theanderose, esculose, and trehalulose [16].

Maltases (EC 3.2.1.20) are classified into glycoside hydrolase family 13 (GH13) along with many other enzymes reacting on α-glucoside linkages, such as α-amylase (EC 3.2.1.1), oligo-1,6-glucosidase/isomaltase (EC 3.2.1.10), and sucrose isomerase/isomaltulose synthase (EC 5.4.99.11) [22]. All GH13 enzymes share a common (β/α)_8_-barrel fold, also called a TIM-barrel. The TIM-barrel is folded from eight parallel β-strands forming a core of the protein that is surrounded by eight α-helices [23]. In 2010, Yamamoto et al. [24] published the first 3D structure of a yeast isomaltase—the IMA1 of *Saccharomyces cerevisiae* (*Sc*IMA1)—that was crystallized in complex with its competitive inhibitor maltose. The *Sc*IMA1 structure constituted of three domains: A, B, and C [24]. Domain A consisted of the (β/α)_8_ barrel common to GH13 enzymes. Domain A contained the catalytic residues Asp215, Glu277, and Asp352 at the C-terminal side of the barrel. Domain B had a loop-rich structure containing one short helix and a small antiparallel β-sheet while the folding of domain C was atypical for GH13 enzymes consisting of five antiparallel β-strands in a double Greek-key motif. The active-site cleft was created by domains A and B forming a pocket-shaped structure as in other GH13 enzymes [23,24]. As the 3D positioning of the three catalytic amino acids is very conserved among the GH13 enzymes [23], substrate specificity of the enzyme (existing for example between the isomaltases and maltases) should be caused by the other determinants on the structure and openness of the active site as suggested earlier [24,25].

Here, we present the first crystal structure of a yeast maltase—the *Ba*AG2. Although numerous maltases, isomaltases, and maltase-isomaltases of yeast origin have been biochemically characterized, only one enzyme of this group—the isomaltase IMA1 from *S. cerevisiae* is structurally determined [24,26]. We compare these two structures with those of bacterial and insect maltases and discuss determinants of substrate specificity of these α-glucosidases. In addition to structural data, we provide novel information on transglycosylation activity of *Ba*AG2 and perform in-depth analysis of reaction products from sucrose synthesized by the *Ba*AG2.

## 2. Materials and Methods

### 2.1. Protein Expression and Purification

The maltase gene of *Blastobotrys* (*Arxula*) *adeninivorans* CBS 8244 (GenBank accession: MZ467078) was amplified from genomic DNA, as described previously in [1]. The *B. adeninivorans* strain was kindly provided by Assoc. Prof. V. Passoth (SLU, Uppsala, Sweden). The plasmid pURI3-BaAG2Cter was expressed in *Escherichia coli* BL21 (DE3) strain [1,27] grown in LB medium supplemented with ampicillin (100 μg/mL). Cells were grown on 37 °C until the optical density at 600 nm reached 0.5. Protein over-expression was initiated by adding 0.5 mM isopropyl thio-β-D-galactoside and cells were further incubated on a shaker 20 h at 20 °C. *E. coli* cells were harvested by centrifugation (4424× *g*, 15 min) at 4 °C and flash-frozen for further purification steps.

For transglycosylation and study of the product spectrum the *Ba*AG2 was purified by immobilized metal ion affinity chromatography (IMAC) as in [1] without further polishing steps.

For crystallization trials, *Ba*AG2 was purified by IMAC followed by size-exclusion chromatography (SEC). Thawing and resuspension of frozen *E. coli* cells were performed in IMAC binding buffer (BB) (20 mM MES, pH 6.5, 300 mM NaCl, 20 mM imidazole) and disrupted with ultrasonication (Q500 Sonicator, QSonica, Newtown, CT, USA). Lysate was cleared by centrifugation (64,000× *g*, 40 min, 4 °C), filtered by a polyethersulfone filter (pore size 0.2 µm, SARSTEDT AG & Co. KG, Nümbrecht, Germany) and applied to His60 Ni Superflow Resin (Takara Bio Inc., Shiga, Japan) using 20 mL Econo-Pac Chromatography Column (Bio-Rad Laboratories, Hercules, CA, USA). The column was washed with 60 mL of BB and maltase was eluted with 20 mL of elution buffer (20 mM MES, pH 6.5, 300 mM NaCl, 300 mM imidazole). The *Ba*AG2-containing fractions were pooled and applied to a SEC column HiLoad Superdex200 16/600 pg (GE Healthcare, Uppsala, Sweden) equilibrated with SEC buffer (20 mM MES, pH 6.5, 150 mM NaCl). With the flow rate of 1 mL/min fractions of 4 mL were collected and analyzed on 10% SDS-PAGE. The peak fractions were pooled and concentrated with Amicon Ultra-15 Centrifugal Filters (30 kDa molecular weight cut-off, MWCO; Merck KGaA, Darmstadt, Germany). Concentrated sample (13.1 mg/mL) was used in further crystallization trials.

### 2.2. Crystallization of the Maltase

*Ba*AG2 with the concentration of 13.1 mg/mL in SEC buffer was used in initial crystallization trials using sitting-drop vapor-diffusion method. Prior to crystallization trials, the protein solution was centrifuged (16,110× *g*, 20 min, 4 °C) to remove any precipitate. Then, 96-well plates were filled with 80 μL of MIDASplus (Molecular Dimensions, Sheffield, UK) or Structure Screen (Molecular Dimensions, Sheffield, UK) or Salt RX (Hampton Research, Aliso Viejo, CA, USA) screen solutions. Sitting drops of 200 nL with ratio 1:1 (protein:crystallization reagent) were set up by Mosquito 9B crystallization robot (SPT Labtech Ltd., Melbourn, UK) at 18 °C.

After a week, needle-like crystals were detected in 0.1 M MES (pH 5.5) supplemented with 12% polyvinylpyrrolidone (PVP) (condition #2 in MIDASplus, Molecular Dimensions, Sheffield, UK. Optimization by varying pH and PVP concentrations was carried out on 48-well sitting-drop plates with 300 μL of reservoir solution and 1 + 1 μL drops with maltase protein (12.7 mg/mL) plus crystallization solution. Seeding was performed by crushing needle-like crystals from the original MIDASplus screen and streaking the seed mixture through the drops with a cat whisker. Plate-like *Ba*AG2 crystals appeared after streak-seeding and two weeks of growth in the original conditions.

The crystals were transferred to a solution containing 0.1 M MES (pH 5.5), 30% PVP with 100 mM α-acarbose (Merck KGaA, Darmstadt, Germany) or D-glucose (Carl Roth GmbH, Karlsruhe, Germany) and incubated overnight at 18 °C. The crystals were then flash-cooled in liquid nitrogen.

### 2.3. Data Collection and Structure Determination

Diffraction data were collected remotely on beamline BioMAX at MAX IV Laboratory (Lund, Sweden) [28]. *Ba*AG2 data with acarbose and glucose in its active center were collected to 2.12 and 2.13 Å resolution, respectively. Data was automatically processed in ISPyB [29] with EDNA software [30] and manually scaled with XDS package [31] using XSCALE and XDSCONV programs. HHpred [32] was used to create a sequence alignment based on the protein sequence of *Ba*AG2 as an input. The best match was the isomaltase IMA1 from *S. cerevisiae* (PDB: 3AJ7), which was used as an input to Sculptor [33] for creating a model for MR that was performed with Phaser [34] in the PHENIX software suite [35]. Firstly, the structure of *Ba*AG2 with acarbose was solved. Next, refined *Ba*AG2-acarbose was used as an input model in rigid-body refinement to solve the *Ba*AG2-glucose structure.

Subsequent refinement was performed in phenix.refine [36]. Ion, ligand placement and manual building of the structure was done in *Coot* [37]. Structure validation was performed using MolProbity [38]. Molecular graphics were prepared using PyMOL [39]. Superimposition of the structures and calculation of the root mean square deviation (r.m.s.d.) between atomic coordinates were carried out using Chimera Matchmaker [40]. The final structures of *Ba*AG2 were deposited to the Protein Data Bank (PDB) [41] with accession codes: 7P01 (*Ba*AG2-acarbose) and 7P07 (*Ba*AG2-glucose).

### 2.4. Transglycosylation Reaction and Product Separation

Transglycosylation reaction with sucrose was conducted similarly as in [1] using 20 µg/mL *Ba*AG2 in 100 mM K-phosphate buffer (pH 6.5) supplemented with 0.2 g/L Na-azide and 500 mM (171.1 g/L) or 1500 mM (513.3 g/L) sucrose at 30 °C up to 72 h. The mixture contained 8 activity units (U) per 1 mL which was determined by using 1 mM *p*-nitrophenyl-α-D-glucopyranoside (*p*NPG) as a substrate. The reaction mixtures also contained 5 mg/mL bovine serum albumin (BSA, fraction V; Amresco, VWR Life Science, Radnor, PA, USA) to aid enzyme stability. Samples were withdrawn at certain intervals for thin layer chromatography (TLC) and high-performance liquid chromatography (HPLC) analysis and heated for 5 min at 95 °C to stop the reaction. Samples were stored at −20 °C. Substrate concentration in transglycosylation assay on turanose and maltulose was 500 mM. The glucose content was determined spectrophotometrically as in [1] in the transglycosylation samples with turanose and maltulose.

The reaction mixtures for product separation and their further detection by nuclear magnetic resonance (NMR) were prepared as shown above and incubated for 24 h. The mixtures were heat-treated (95 °C, 5 min) and diluted in sterile mQ water to contain approximately 10 mg/mL of sugars. Protein was removed by filtration using Vivaspin 500 Centrifugal Concentrators with 10 kDa MWCO (Sartorius Lab Instruments GmbH & Co. KG, Goettingen, Germany) and 0.5 mL of the sample was loaded onto a gel filtration column system constructed of 3 SEC columns of HiLoad 16/600 Superdex 30 pg (GE Healthcare, Uppsala, Sweden) in series. Sugars of different degree of polymerization (DP) were separated by SEC on ÄKTA prime plus chromatography system (GE Healthcare, Uppsala, Sweden) at room temperature using filtered and degassed mQ water with the flow rate 0.1 mL/min. One mL fractions were collected, analyzed for the presence of reducing sugars similarly as in [42], and applied TLC (see Section 2.6). Reducing sugars were detected by combining 32 µL of fraction, 64 µL of DNSA reagent, heating the sample 5 min 100 °C and adding 128 µL of mQ water. Then, 200 µL of the sample was transferred to 96-well flat-bottom transparent polystyrene microplate well (Greiner Bio-One, Frickenhausen, Germany) and optical density at 540 nm was measured by using Infinite M200 PRO microplate reader (Tecan Group Ltd., Männedorf, Switzerland). The fractions containing sugars of DP 2 or DP 3 were separately pooled from two identical runs and dried at 40 °C using SpeedVac concentrator (Thermo Fisher Scientific, Waltham, MA, USA). The purity of DP 2 and DP 3 samples was confirmed by TLC analysis. The DP 2 and DP 3 samples were separately dissolved in 0.5 mL D_2_O (Merck KGaA, Darmstadt, Germany) and subjected to NMR analysis.

### 2.5. Nuclear Magnetic Resonance (NMR)

NMR spectra were recorded at 25 °C on a Bruker Avance III (799.88 MHz for ^1^H and 201.14 MHz for ^13^C) (Bruker Corp., Billerica, MA, USA) equipped with a 5 mm TCI ^1^H/(^13^C, ^15^N) cryoprobe. The mixing time for heteronuclear single quantum coherence-total correlation spectroscopy (HSQC-TOCSY) measurements were 60 ms. Heteronuclear multiple bond correlation (HMBC) was optimized for 10 Hz long range coupling constants. The narrow HSQC spectrum was acquired using 256 F1 increments with a F1 spectral width of 16 ppm and the transmitter offset set to 95 ppm, as opposed to 256 F1 increments with a spectral width of 165 ppm and transmitter offset at 75 ppm in the traditional HSQC. All two-dimensional spectra were recorded using standard Bruker pulse sequences, acquired using Bruker TopSpin software (ver. 3.5), and processed using TopSpin (ver. 4.0). Sucrose, maltose, melezitose, isomaltulose, maltulose, panose, and turanose were used to obtain reference spectra.

### 2.6. Thin Layer Chromatography (TLC) and High Performance Liquid Chromatography (HPLC)

Hydrolysis and polymerization products of sucrose by *Ba*AG2 were visualized using the TLC analysis as in [1,43] on Silica Gel 60 F_254_ plates with a concentrating zone (Merck KGaA, Darmstadt, Germany). Then, 0.5 µL of the reaction mixtures or concentrated fractions from SEC were spotted onto the plate and sugars were separated with two runs in chloroform:acetic acid:water (6:7:1, v:v:v) as in [44]. If needed, the reaction mixtures were appropriately (2 or 5 times) diluted in mQ water. Sugars were visualized by spraying the plates with aniline-diphenylamine reagent and subsequent heating of the dried plates at 100 °C [1,45]. Fructose, glucose, maltose, sucrose, trehalulose, isomaltose, isomaltulose (palatinose), turanose, maltulose, maltotriose, panose, erlose, isomelezitose, and melezitose were used as reference sugars. Maltulose, trehalulose and erlose were bought from Carbosynth Ltd. (Compton, Berkshire, UK), panose was from Hayashibara Co. Ltd. (Okayama, Japan) and palatinose from BENEO-Palatinit GmbH (Mannheim, Germany). Isomelezitose was obtained as shown in [1]. Other saccharides were from Sigma-Aldrich (Merck KGaA, Darmstadt, Germany).

HPLC analysis was performed as in [1,46]. Glucose and fructose were used to calibrate the Alltech Prevail Carbohydrate ES column (Grace, Deerfield, IL, USA).

### 2.7. Alignment of Protein Sequences

Protein sequences were retrieved from the GenBank and the PDB and were aligned using Clustal Omega [47]. Genome of the *B. adeninivorans* strain TMCC 70007 in the GenBank (JACADK000000000.1; assembly accession: GCA_016162255.1) was searched with the BLAST tool using the *Ba*AG2 gene sequence (GenBank: MZ467078) as a bait to retrieve a *Ba*AG2 homologue from the genome.

## 3. Results

### 3.1. In Silico Analysis of Maltase Sequences from Blastobotrys Species

Previously we have cloned a maltase gene (GenBank accession: MZ467078) from the type strain of *B. adeninivorans* CBS 8244, produced the *Ba*AG2 protein by heterologous expression and conducted its biochemical characterization [1]. The substrates of *Ba*AG2 are shown in Appendix A. Currently there are two whole genome sequences of *Blastobotrys* species available: the industrial strain LS3 recently reclassified as belonging to *B. raffinosifermentans* [2,4] and the *B. adeninivorans* strain TMCC 70007 newly isolated from the Pu-erh tea fermentation [48]. From the scaffold 1 (JACADK010000001.1; 2254288–2256027) of the TMCC 70007 genome we disclosed a gene with a high identity (97% at nucleotide sequence level) to the gene encoding the *Ba*AG2. Comparison of the *Ba*AG2 amino acid sequence with its two homologues from the strains LS3 and TMCC 70007 showed that the CBS 8244-originating *Ba*AG2 had the highest identity (97.59%) to putative maltase of TMCC 70007 and slightly lower identity (95.86%) to respective homologue in LS3. The identity between the two predicted maltases from strains TMCC 70007 and LS3 was 95.96%. Clustal Omega alignment of these three protein sequences is presented in Appendix A (Appendix A). All amino acids that either participate directly in the catalysis (catalytic triad) or are involved in the binding of substrates to the enzyme (see also Figure 1 and Figure 2), are fully conserved between all three sequences.

### 3.2. Overall Quality and General Characterization of the BaAG2 Maltase Structure

The crystal structure of the *Ba*AG2-acarbose complex was solved at the resolution of 2.12 Å using the molecular replacement method with the isomaltase IMA1 (PDB: 3AJ7) of *S. cerevisiae* applied as a search model. The structure of the *Ba*AG2-glucose complex was solved at the resolution of 2.13 Å using the rigid-body refinement against the *Ba*AG2 maltase-acarbose model. Statistics of data processing and structure refinement are shown in Appendix A.

The crystals belonged to the space group P 1 2_1_ 1 and contained two molecules in the asymmetric unit. Size-exclusion chromatography indicated that *Ba*AG2 is most likely monomeric in solution (Appendix A). GH13-family enzymes have been identified as monomers or dimers. Crystallized silkworm sucrose hydrolase, an exo-acting α-glucosidase of GH13, was confirmed as a dimer [50], also amylases of GH13 family were dimeric [51]. In contrast, IMA1 of *S. cerevisiae* and GH13_31 α-glucosidase of *Bacillus* sp. have been described as monomers [24,52].

In general, the overall structure of *Ba*AG2 is similar to that of bacterial maltases [52,53], isomaltase of *S. cerevisiae* [24,26], bacterial dextran glucosidase [54], and α-amylases of different origin [55,56,57]. The structure of *Ba*AG2 is composed of four domains. Domain A is a typical (β/α)_8_ or TIM-barrel (residues 1–110, 191–386, 485–505), formed by eight parallel β-strands, surrounded by eight α-helices (Figure 1a). Domain A also harbors the catalytic triad at the C-terminal side of the barrel: a nucleophile Asp216, a general acid/base catalyst Glu274 and Asp348 as a stabilizer of the substrate to accelerate the catalysis (Figure 1b). Domain B (residues 111–190) is a loop-rich structure containing a helix and an antiparallel β-sheet, while domain B’ (residues 387–484) comprises a loop- and helix-rich region inserted between the eighth β-strand and α-helix (β-α_8_). Domain C (residues 506–581) is formed of seven antiparallel β-strands in a double Greek-key motif.

An electron density indicating a metal ion was visible in both molecules. The metal ion has an octahedral geometry and it is coordinated by oxygen atoms of N-terminal amino acids: Asp32 OD1, Asn34 OD1, Asp36 OD1, Ile38 O, Asp40 OD2, and with one water molecule. The metal ion was modelled as Ca^2+^ ion due to the 2.4 Å coordination distances [58]. The Ca^2+^ ion was bound in the cavity formed by the loop just before the first α-helix of domain A. Similar metal binding is visible also in structures of other α-glucosidases [24,52,53]. As no additional metal ions (except NaCl) were used in buffers during protein purification and crystallization trials, the Ca^2+^ ion could possibly originate from the LB medium, which was used for growing of maltase-expressing *E. coli* cells.

The effect of 1 mM EDTA and 1 mM CaCl_2_ to the activity of *Ba*AG2 was evaluated spectrophotometrically using 1 mM *p*NPG as a substrate as in [1] and the differences in activities with or without adducts were negligible. A similar result has been obtained for an α-glucosidase from fungus *Chaetomium thermophilum* var. *coprophilum* [59]. However, a slight activation of 5 mM Ca^2+^ has been shown for two (intracellular and extracellular) α-glucosidases from *Brettanomyces bruxellensis* (previously *Br. lambicus*) [60]. We consider that the presence of Ca^2+^ ion is not vital for the activity of *Ba*AG2 at optimal conditions but may have a stabilizing effect at some specific environments. Ca^2+^ ions often have a crucial role in maintaining structure and stability of α-amylases and dextran glycosidases, especially thermophilic ones or at elevated temperatures [61,62]. However, it has to be noted that α-amylases have several binding sites for Ca^2+^ ions in their structure [61] whereas in the case of *Ba*AG2 only one site was disclosed (Figure 1a).

The r.m.s.d. value was calculated with Chimera matchmaker [40] to compare the *Ba*AG2-acarbose and *Ba*AG2-glucose structures. The best aligned pair of chains were Chain B from maltase-acarbose structure and Chain A of maltase-glucose structure. R.m.s.d. values between the main chain and all atoms were respectively 0.418 Å and 0.445 Å. Both ligands, acarbose, and glucose, are bound strongly to −1 subsite by Asp71, Tyr74, His114, Arg214, Asp216, Glu274, His347, Asp348, and Arg440. The main difference between the structures of *Ba*AG2-acarbose and -glucose is the orientation of the Asp216 side chain, which acts as a nucleophile in the reaction. On the *Ba*AG2-glucose structure, Asp216 does not reach the anomeric carbon of the glucose and therefore does not contribute to binding of the ligand in the active center (Figure 1b). In the *Ba*AG2-acarbose structure the position of Asp216 has similar orientation as in the case of other structures of α-glucosidases [24,52,53].

Acarbose (unsaturated cyclitol moiety, also named valienol, linked via nitrogen to maltotriose) is a competitive inhibitor of α-glucosidases from GH13 and GH31 families [1,43,63]. It has been used as an active site-binding ligand in crystallization of α-amylases, for example that of *Aspergillus oryzae* (PDB: 7TAA) [64] and of human intestinal maltase-glucoamylase (PDB: 2QMJ) [65]. Importantly, α-glucosidases cannot hydrolyze *N*-glycosidic bond [66] in the molecule and therefore acarbose can be used as a ligand for co-crystallization of a catalytically active enzyme. As acarbose inhibits breakdown of complex carbohydrates into glucose by binding to brush border α-glucosidases and pancreatic α-amylase, it is used as an anti-diabetic drug [67]. The solved structure of *Ba*AG2-acarbose complex reveals the binding mode of α-acarbose to the maltases: it is positioned in the active center cavity with valienol moiety based at the bottom. Valienol is bound to the −1 subsite of *Ba*AG2 and adjacent maltotriose moiety is binding at subsites +1, +2, and +3, showing considerable similarity with the binding mode recorded in domestic silk moth (*Bombyx mori*) sucrose hydrolase *Bm*SUH-acarbose structure (PDB: 6LGE) [50]. Additional electron density is visible for another α-acarbose molecule on *Ba*AG2-acarbose model on chain A. It is bound to the enzyme’s surface and contributes to crystal packing: amino acids from chain A and from symmetry molecule chain A are at H-bond distance away. Contributing amino acids from the symmetry-related molecule are Ser358, Asp359, Ala360, Gln436, Lys437, Asp537 and Asn34, Gly35, and Gly460.

Every refined *Ba*AG2 molecule had an electron density at the surface of *Ba*AG2, which was modelled as a monomer of PVP. The shallow opening was formed by side chains of Ile38, Gln69, Met72, His83, Pro85, Tyr86, and Trp463, while only in the case of His83, the side chain reached the H-bond distance from the PVP monomer. An additional PVP molecule was modelled to chain A of the *Ba*AG2-acarbose structure, where its *N*-vinylpyrrolidone core was positioned in parallel with Tyr505 at the distance of 4 Å.

### 3.3. Characterization of the Transglycosylation Activity of BaAG2 on Sucrose

We have shown a considerable transglycosylating activity of *Ba*AG2 with maltose as a substrate, resulting in the synthesis of trisaccharides—maltotriose and panose, and an α-1,6-linked disaccharide isomaltose [1]. By 24 h of reaction with 500 mM maltose, the amount of transglycosylation products reached up to 22.4 mg/mL constituting 12.6% from total sugars [1]. In current work, we assayed transglycosylating activity of *Ba*AG2 on sucrose.

For preliminary analysis, TLC assay of *Ba*AG2 samples incubated with 250 mM or 500 mM sucrose as a substrate was carried out. Already within 2 h of reaction at least two different trisaccharides and a disaccharide different from sucrose were detected in the reaction samples (Appendix A) confirming the active transglycosylation. HPLC analysis was used to quantify the saccharides in the course of the reaction. Elevated concentration of the substrate resulted in a higher amount of transglycosylation products, especially of DP 3 species (Figure 3). By 24 h-reaction on 500 mM sucrose the amount of transglycosylation products reached 19.1 mg/mL (11.2% of total sugars, *w/w*). The amount of transglycosylation products was 123.4 mg/mL (23.2% of total sugars, *w/w*) where DP 3 products constituted 100.1 mg/mL when substrate concentration was raised to 1500 mM. The residual sucrose content was approximately 50% indicating partial reaction at the used conditions.

Due to the methodological limitations, sucrose isomers turanose, maltulose, palatinose, and trehalulose (see Appendix A) did not fully separate from sucrose using the applied HPLC method, and only a few reference DP 3 saccharides were available to verify the identity of trisaccharides. Therefore, in-depth structural analysis of transglycosylation products was conducted using NMR spectroscopy to verify the chemical composition and linkage types of the di- and trisaccharides (see Section 3.4).

We report here that in addition to sucrose, its α-1,3- and α-1,4-linked isomers turanose and maltulose, respectively, which are hydrolyzed by the *Ba*AG2 [1], also serve as substrates for transglycosylation. Appendix A shows that oligosaccharides with DP 3 are synthesized from turanose and maltulose by *Ba*AG2. The glucose concentration determined by enzymatic assay from 24 h reaction was approximately 17 mg/mL for both substrates, which was considerably lower than in the case of sucrose as a substrate (shown in Figure 3b). This can be explained by the fact that maltulose and turanose are respectively two and three times less efficient substrates for the enzyme compared to sucrose [1].

### 3.4. Identification and Structural Determination of the Transglycosylation Products of Sucrose

Transglycosylation products of samples originating from 24 h-incubation of *Ba*AG2 with 500 mM or 1500 mM sucrose were separated by SEC to fractions of different DP (see Section 2.4). The substrate concentrations were chosen according to earlier recorded efficient transglycosylation on 500 mM maltose by the *Ba*AG2 [1] and sucrose concentration close to 1500 mM (500 g/L = 1460 mM) used in transglycosylation reaction by α-glucosidases from *Metschnikowia* sp. [16,17]. We chose 24 h reaction time as longer incubation caused noticeable decrease in DP 3 reaction products from 500 mM sucrose (Figure 3b).

The fractions with saccharides of different DP were identified based on the TLC analysis and reducing sugar content of the samples. The fractions containing hydrolysis products, i.e., glucose and fructose, were discarded, while fractions containing di- and trisaccharides were separately collected. DP 2 saccharides-containing fractions eluted at retention volume 293–299 mL and DP 3 sugars appeared in fractions at retention volume 281–288 mL. After drying of samples with the same DP from two identical runs, approximately 1.2 mg (DP 2) and 1.0 mg (DP 3) of sugar mixture prepared from a reaction with 500 mM sucrose was obtained. From the reaction with 1500 mM sucrose respective amounts were 5.1 mg and 2.1 mg.

The DP 2 and DP 3 fractions were analyzed using NMR spectroscopy (see Section 2.5) to disclose chemical entities of oligosaccharides produced. In order to determine the relative concentration of each disaccharide, the integrals of non-reducing anomeric glucose HSQC signals were used from the narrow HSQC. The HSQC spectra of the DP 2 and DP 3 samples originating from 500 mM sucrose as substrate are shown in Appendix A, respectively. By using various 2D NMR experiments, as well as comparison to reference compounds and literature data from [17], the different anomeric signals were assigned. Both of the fractions contained several components of various linkage types at different ratios. The proportions of the di- and trisaccharide products prepared using 500 mM or 1500 mM sucrose as a substrate are shown in Table 1 and Table 2, respectively.

A mixture of sucrose and its analogues was detected in the DP 2 fraction. As presented in Figure 3, the most abundant disaccharide in the reaction mixture was sucrose—the unreacted substrate. When sucrose concentration was 1500 mM, by 24 h-reaction, sucrose constituted nearly 70% of the DP 2 fraction, but other disaccharides were still detectable (Table 1). The main α-1,4-linked disaccharide produced during transglycosylation was maltulose, whereas a small proportion of maltose was also produced from glucose used as an acceptor. An α-1,1-linked disaccharide, trehalulose, was the second most abundant product and a considerable proportion of α-1,3-linked sugars, i.e., turanose and nigerose, were also produced (Table 1). The synthesis of a significant amount of trehalulose during transglycosylation of sucrose by yeast α-glucosidases from *Metschnikowia*
*reukaufii* and *M. gruessii* has been reported recently [16]. Importantly, trehalulose was also proved as a substrate hydrolyzed by *Ba*AG2 (Appendix A), widening the known substrate spectrum of the *Ba*AG2. Substrates of *Ba*AG2 identified by us are listed in Appendix A. A small proportion of isomaltose-type linkage was also produced from sucrose by *Ba*AG2 similarly as previously shown for maltose as a substrate [1]. Similarly, to *Ba*AG2, only minor amounts of isomaltose were detected from transglycosylation reaction of maltose by α-glucosidase from *Xanthophyllomyces dendrorhous* [14,68] in contrast to the respective enzyme of *S. cerevisiae* that produced mainly isomaltose [14].

Glucose and fructose are hydrolysis products of sucrose. Our results indicate that one of them—fructose—is a more effective acceptor for *Ba*AG2 in transglycosylation reaction than glucose as most of the synthesized disaccharides had a fructose unit linked to the glucose moiety (Table 1). The hydrolysis of already produced trisaccharides is unlikely during 24 h-reaction as sucrose, four times more effective substrate compared to for example melezitose [1], was far from depletion (Table 1; Figure 3b). Interestingly, some transglycosylation products (for example isomaltose and palatinose) of low abundance contained a linkage that *Ba*AG2 is unable to hydrolyze [1].

Practically no reducing glucose was detected from DP 3 fraction. One of the main DP 3-compounds detected in the reaction mixture was isomelezitose, which accounted for approximately 1/3 of the anomeric signal of the sample from reaction with 500 mM sucrose (Table 2). The second most abundant trisaccharide in this sample was identified as erlose, which was the most abundant sugar among the products resulting from reaction with 1500 mM sucrose. *Ba*AG2 hydrolyzed erlose when it was provided as a sole substrate (Appendix A) but hydrolysis of the trisaccharide during the transglycosylation reaction is most likely not significant due to the abundant presence of sucrose.

In addition, a significant proportion of theanderose was detected from both DP 3 samples. Elevated concentration of sucrose enhanced the formation of melezitose, but hindered the production of α-1,3-linked esculose (Table 2). In accordance with our results, a significant amount of isomelezitose and detectable portions of erlose and theanderose were produced from sucrose by transglycosylating α-glucosidases from *M. reukaufii* and *M. gruessii* [16,17].

1-O^F^-glucosyl-sucrose (Glc-α-1,1-Fru-β-2,1-Glc) was identified among products formed from sucrose by *Ba*AG2 (Table 2; NMR assignment shown in Appendix A). According to our knowledge this is the first report of the trisaccharide detected among transglycosylation products of maltases and it has not previously been assigned using NMR spectroscopy. Interestingly, a previously uncharacterized saccharide which appears to arise from further transglycosylation of turanose (Glc-α1,4-Glc-α1,3-Fru) was also disclosed (Table 2).

## 4. Discussion

### 4.1. The Architecture of the Substrate-Binding Pocket of BaAG2 Reveals the Binding Mode of the Ligand and Exhibits Water Channels

In this study we solved two ligand-bound crystal structures of the maltase *Ba*AG2 from an early-diverged yeast *B. adeninivorans*. Crystal soaking enabled us to bind acarbose or glucose into the active center of the catalytically active maltase. A more detailed view of ligand binding is visualized in Figure 4. We have previously predicted Asp216, Glu274, and Asp348 as the catalytic triad residues of *Ba*AG2 [1] and Figure 4 shows the side chains of these residues are in close vicinity of the ligands.

We compared *Ba*AG2 amino acid sequence with four characterized α-glucosidases: the bacterial maltase BspAG13_31 of *Bacillus* sp., the α-glucosidase (maltase) from a floral nectaries colonising yeast *M. reukaufii,* the isomaltase IMA1 of *S. cerevisiae* and the maltase-isomaltase *Op*Mal1 of a non-conventional yeast *O. polymorpha* (Figure 2). Identity values between the protein sequences were moderate, mostly from 42 to 49%. The highest identity (63%) was detected between the sequences from *M. reukaufii* and *O. polymorpha.* Identity of the bacterial maltase to yeast enzymes ranged from 42 to 45%. According to CAZy database [69] maltases with the highest identity to *Ba*AG2 (i.e., *Mr*AG, *Op*Mal1, *Ao*MalT), belong to GH13 subfamily 40 (GH13_40), harboring several fungal enzymes including yeast maltases and isomaltases. The only structure solved for GH13_40 thus far was isomaltase *Sc*IMA1.

Figure 2 shows that the residues binding the substrate at −1 subsite (shown on yellow background on the alignment) are fully conserved between the selected enzymes regardless of their substrate preference. Already 20 years ago, MacGregor et al. stated a high conservation of amino acids at −1 subsite of α-amylase-family enzymes by comparing their sequences and structures. The amino acid residues building up subsites +1 and further “+” subsites were expected to vary due to the specificity of the enzyme [49].

The structure of *Ba*AG2-acarbose complex reveals a possible binding mode of the substrate to the enzyme. The −1 subsite is lined by side chains of Asp71, Tyr74, His114, Arg214, Asp216, Glu274, His347, Asp348, Arg440, and one water molecule interacting with the valienol moiety (Figure 4a). At +1 subsite only Glu274 is stabilizing the ligand, assisted by three water molecules, which are mainly interacting with other water molecules in the vicinity. Ligand residue at +2 subsite is stabilized by the side chain of Glu309 along with seven water molecules in the close proximity. The most distant residue of acarbose is positioned to subsite +3, where it makes contacts with the side chain of Arg233 and four water molecules (Figure 4a). Positions of these both amino acids, Glu309 and Arg233, responsible of ligand binding in the further subsites do not align well with either *Sc*IMA1 of *S. cerevisiae* nor BspAG13_31 structures (see Figure 5f). Similar binding mode of the ligand is observed in the case of *Ba*AG2-glucose complex, where glucose is positioned to −1 subsite, providing contacts with side chains of Asp71, His114, Arg214, Glu274, His347, Asp348, and Arg440, and with one water molecule. The main noticeable difference of ligand binding at −1 subsite between the two structures is the orientation of the side chain of Asp216 which seems to be turned away from the C1 atom of glucose (Figure 1 and Figure 4) in the case of *Ba*AG2-glucose. The conformation where the catalytic nucleophile points away from the C1 atom of glucose have been recorded recently for silkworm sucrose hydrolase *Bm*SUH [50]. The authors report on novel finding of conformational changes in the active site during the catalytic cycle of *Bm*SUH where closed configuration changed the direction of nucleophile side chain [50]. So, the conformational flexibility of the side chain of Asp216 of *Ba*AG2 with either glucose or acarbose as a ligand can be explained as reflection of different stages of the catalytic cycle.

The active site clefts of maltases and isomaltases have evolved to accommodate only maltose-like (in the case of maltases) or isomaltose-like (isomaltases) sugars. In addition, there are enzymes with wide substrate specificity called maltases-isomaltases that can bind and hydrolyze both types of substrates [43]. We will address structural aspects of substrate preference in the Section 4.2.

Importantly, for the complete hydrolysis of the substrate, water molecules are essential. Structural study of the isomaltase IMA1 of *S. cerevisiae* revealed that as the shape of the active site entrance of the enzyme cannot accommodate both the incoming substrate and outgoing water [24], extra water channels or reservoirs are needed for the reaction. Yamamoto et al. (2010) described two chains of water molecules in the structure of the *Sc*IMA1 and proposed that the incoming substrate is pushing out water molecules from the active center through the channels [24]. We have observed similarly positioned chains of water molecules in the structure of *Ba*AG2 (Figure 4c). The first chain comprises three water molecules: W3, W14, and W47, where W47 is the closest to acarbose at −1 subsite. It also forms a direct hydrogen bond to Asp71 OD1 and carbonyl oxygen of its main chain and to Arg440 NH1 and Arg444 NH2. W3 gives contacts to Asp71 OD1, Asp407 OD1, and Arg444 NH1. Finally, W14 is reaching the surface of *Ba*AG2 and gives hydrogen bonds with the main chain carbonyl oxygen of Val70 and nitrogen of Val408. The distances between the water molecules are 3.2 Å. Similar water chain is observed in the case of structures of oligo-1,6-glucosidase of *B. cereus* [70], dextran-glucosidase of *S. mutans* [54], amylosucrase from *Neisseria polysaccharea* [71], sucrose phosphorylase from *Bifidobacterium adolescentis* [72] and silkworm sucrose hydrolase *Bm*SUH [50] suggesting the universal structural mode for α-glucosidases with various activities and origin.

Additionally, a water reservoir is observed in the cleft of the bottom of the active center (W98, W106, W109, W126, and W657 according to *Ba*AG2-acarbose structure numbering), where acid-base catalyst Glu274 and stabilizer Asp348 are making polar contacts with the first water molecule W126 (Figure 4c). The distances from the W126 to Glu274 OE2 and Asp348 OD2 are 2.5 and 2.9 Å, respectively, and it may participate in the hydrolysis. Next, water W109 from that chain is 2.9 Å apart and creating direct hydrogen bonds with Arg214 NH2. Water W98 between W109 and W106 with the distance of 2.9 Å and 2.7 Å, respectively. W98 creates contacts with Asn346 ND2. W106 is before the last in the line—W657, with the distance between them measured at 2.5 Å. W106 is coordinated with Glu345 carbonyl oxygen of its main chain. The last water in the chain W657 is within hydrogen bond distance from Gln24 OE1 and carbonyl oxygen of Leu385 main chain. The chain of waters is lined with hydrophobic amino acids such as Trp60, Phe298, Phe300, Phe343, Try384, and Tyr386, positions of which are overlapping with that of hydrophobic amino acids Trp58, Phe301, Phe303, Tyr347, Tyr387, and Tyr389 in the isomaltase IMA1 structure [24]. Analysis of the structures and literature data indicated that water channels of *Ba*AG2, isomaltase *Sc*IMA1 [24], α-glucosidase BspAG13_31 from *Bacillus* sp. [52] and barley α-amylase [73] are similar.

### 4.2. The Active Center of the BaAG2 Is Similar to That of the Bacterial Maltase, but Differs from Enzymes Specialized to Hydrolyze Isomaltose-like Sugars

Comparison of the structures of the maltases: the *Ba*AG2 of a yeast *B. adeninivorans* and maltase BspAG13_31 of bacterium *Bacillus* sp. AHU2216 with the structure of the isomaltase IMA1 of a yeast *S. cerevisiae*, revealed similar accommodation mode of the substrate at −1 subsite while at +1 subsite accommodation modes of the substrate between the two maltases and the isomaltase were rather different. In all three enzymes, glucose is positioned in the −1 subsite (valienol in case of acarbose as a ligand) and binding at this subsite is the tightest—mediated by numerous contacts. Amino acids contributing to substrate binding at −1 subsite of the *Ba*AG2 (Figure 4a) are overlapping and positioned perfectly in line when compared with respective amino acids of the bacterial maltase BspAG13_31 and the yeast isomaltase IMA1, except for introduced mutations: the acid base catalyst Glu256Gln in BspAG13_31 (PDB: 5ZCE) to bind its native substrate maltotetraose, and Glu277Ala in *Sc*IMA1 (PDB: 3AXH) to bind its native substrate isomaltose [26,52]. In +1 subsite the positioning of glucose correlates very well between the structures of the two maltases: *Ba*AG2 and BspAG13_31 (Figure 6a), despite acarbose in the *Ba*AG2 structure that creates polar contacts only with Glu274 in +1 subsite. In the case of BspAG13_31, glucose residue of maltotetraose at +1 subsite is bound by several amino acids: His203, Glu256, and Gln258 [52]. At binding of isomaltose to the isomaltase *Sc*IMA1, the glucose residue at +1 subsite has a different positioning (Figure 6), with amino acids such as Glu411 and Arg442 contribution to binding [24,26].

According to Voordeckers et al. (2012) and Yamamoto et al. (2011) one of the key residues determining the substrate specificity of *S**c*IMA1 is Gln279 [26,74]. This residue is blocking the binding of maltose-like sugars to the enzyme at +1 subsite. Equivalent amino acid of the maltase BspAG13_31 is Asn258 (see Figure 2), which has a shorter side chain than glutamine and it allows accommodation of maltose-like sugars in the active center (Figure 5a). According to Auiewiriyanukul et al. [52], Asn258 is one of the substrate specificity determinants of this bacterial maltase. In the *Ba*AG2 structure, the loop harboring corresponding amino acid is turned more towards the interior of the protein, enlarging the space for the binding of the substrate at +1 subsite. It has to be noted that this region of *Ba*AG2 is rather different from respective region of *Sc*IMA1.

Another amino acid proposed to determine the substrate specificity of α-glucosidases is Val216 in IMA1 of *S. cerevisiae* [26,74,75]. This bulky hydrophobic amino acid is conserved in sequences of enzymes hydrolyzing α-1,6-glucosidic linkage [26] while yeast enzymes capable of α-1,4-glucosidic linkage hydrolysis have a Thr at respective position [1,43,75,76]. Figure 2 shows that the *Ba*AG2, the maltase of *M. reukaufii* and the maltase-isomaltase Mal1 of *O. polymorpha* have a Thr at the position corresponding to Val216 of the *Sc*IMA1. However, the bacterial maltase BspAG13_31, which is very specific for maltose and short malto-oligosaccharides, and almost unable to hydrolyze sucrose and isomaltose, has an Ala at a respective position (Figure 2 and Figure 5b). Literature data show that bacterial α-glucosidases specific for maltose often possess an Ala at the above-mentioned position [77,78]. Notably, substitution of Ala with Val at the respective position in the *Bacillus stearothermophilus* α-1,4-glucosidase (maltase), abolished its maltose-hydrolyzing activity whereas substitution of Val with Ala in the α-1,6-glucosidase (isomaltase) of the *B. thermoglucosidasius* caused an appearance of maltase activity and a major reduction in isomaltase activity [77].

When the structures of *Ba*AG2 and *Sc*IMA1 are superposed, the side chains of valine and threonine are positioned perpendicularly (Figure 5b), and the side chain of valine is hindering the binding of maltose to the active center of the isomaltase [26]. Mutational analysis has confirmed significance of these residues in substrate specificity of yeast α-glucosidases. Hence, when Val216 was substituted in *Sc*IMA1 with a Thr, the enzyme gained the ability to hydrolyze maltose [75]. Reverse mutation was introduced to maltase-isomaltase Mal1 of a methylotrophic yeast *O. polymorpha* [43]. This enzyme can hydrolyze both maltose and isomaltose, thus representing an α-glucosidase in which both maltase and isomaltase activities are well optimized in a single enzyme. Substitution in *Op*Mal1 of Thr200 (corresponds to Val216 of *Sc*IMA1) with Val drastically reduced the hydrolysis of maltose-like substrates, confirming the requirement of Thr at this position to enable this function. This position has been used to predict the ability of an enzyme to hydrolyze maltose [74,76]. So, a Thr is present at the position corresponding to Val216 of the *S. cerevisiae* IMA1 not only in yeast maltases, but also in enzymes hydrolyzing both maltose and isomaltose, for example in maltase-isomaltases of *O. polymorpha* and *Scheffersomyces stipitis* [43,76].

Positively charged His with a large side chain at the position 203 in BspAG13_31 at +1 subsite (see Figure 5c) narrows the active center, allowing it to accommodate only maltose-like sugars. Interestingly *Ba*AG2 has an isoleucine (Ile220) at this position, while *Sc*IMA1 has a leucine (Leu219) (see Figure 2 and Figure 5c). Ile220 and Leu219 are not filling the same space as His203 does in BspAG13_31. His203 in BspAG13_31 structure together with Ala200 and Asn258 are involved in the recognition of malto-oligosaccharides as substrates and are considered as structural determinants for specificity of GH13_31 enzymes towards α1,4-glucosidic linkage [52]. Additionally, the shape of the active center of *Ba*AG2 is modified by Phe300, pointing to the +1 subsite, where it creates a hydrophobic surface for the binding of the substrate (Figure 5d). The *Sc*IMA1 structure also has a phenylalanine (Phe303) at the same location—it pushes the active center cavity towards the reducing glucose residue of the ligand—isomaltose.

The mouth of the active center cavity of *Sc*IMA1 is bordered by a loop Val308-Glu322 [24]. In this loop, Arg315 is pointing to +2 subsite preventing binding of maltose-like sugars [24]. At the same position of corresponding loop, Leu305-Met319, the *Ba*AG2 has a glutamate (Glu309), side chain of which is pointed to the opposite direction compared to that of Arg315 in *Sc*IMA1 and it is stabilizing acarbose molecule at +2 subsite (Figure 5e). The corresponding loop is not well-defined on the structure of BspAG13_31 [52], making it difficult to compare with the loop in yeast α-glucosidases.

To compare how different sugars would fit to the active center of the *Ba*AG2, sucrose, maltose and isomaltose from ligand-bound structures of different GH13 family enzymes were superimposed (Figure 7). We aligned *Ba*AG2-acarbose structure with that of sucrose hydrolase *Bm*SUH (PDB: 6LGF) from domestic silk moth *B. mori* to superimpose sucrose and acarbose molecules. Glucose (from sucrose) and valienol (from acarbose) moieties were aligning perfectly on top of each other at −1 subsite (Figure 7a), but at +1 subsite the furanose ring of fructose residue from sucrose was positioned perpendicularly to the pyranose ring of glucose from the acarbose. Still, sucrose on the *Ba*AG2-acarbose model did not clash with amino acids lining the surface of *Ba*AG2 active center.

Superimposing the acarbose from the *Ba*AG2-acarbose and maltose from the BspAG13_31 maltose-bound structure revealed how acarbose was mimicking binding of maltose to the active center of maltases (Figure 7b). Glucose moiety at −1 subsite was perfectly aligned with the valienol group of acarbose. If isomaltose from *Sc*IMA1 was superimposed to acarbose of *Ba*AG2, a completely different orientation of glucose moieties at +1 subsite was observed. Glucose residue from isomaltose (at +1 subsite) was oriented perpendicularly with the glucose residue from the acarbose molecule (Figure 7c). Still, isomaltose and sucrose fitted to the active center of the *Ba*AG2 because the amino acids lining the active center cavity did not narrow it down. Similarly to acarbose, isomaltose is a competitive inhibitor of *Ba*AG2 [1]—it can bind to the active site but further hydrolysis is blocked.

### 4.3. In Transglycosylation Reaction of BaAG2 on Sucrose a Variety of Bonds Are Synthesized

The transglycosylation ability has been previously reported for several fungal α-glucosidases. For example, α-glucosidase of *X. dendrorhous* [14], AgdB of *A. nidulans* [15], MAL62 of *S. cerevisiae* [1], as well as *Ba*AG2 [1] are able to use maltose as a substrate to produce (iso)malto-oligosaccharides as transglycosylation products. Transglycosylation of sucrose has been much less studied among α-glucosidases, most likely due to the fact that (i) GH31 maltases are not able to hydrolyze sucrose [14] and (ii) separation and in-depth analysis of some sucrose transglycosylation products is complicated. The first report on transglycosylation of sucrose by the *S. cerevisiae* maltase producing isomelezitose is from 1979 [21], and in this work only a minor amount of this trisaccharide (0.4% from total sugars) was detected. Recent studies on two α-glucosidases from *Metschnikowia* ssp. show a strong transglycosylation activity on sucrose and formation of a mixture of rare sugars, including isomelezitose, erlose, theanderose, esculose, and trehalulose—the sugars also found in honey. The isomelezitose amounts obtained in reactions mediated by the recombinant proteins, ~ 170 g/L, were the highest reported so far [16,17].

Sucrose (Glc-α-1,2-Fru) is one of the best substrates among the ones tested for the maltase *Ba*AG2 of *B. adeninivorans* [1], therefore it was used as a substrate for transglycosylation reactions. Our results on *Ba*AG2 indicated significant transglycosylation ability, especially on 1500 mM sucrose (see Section 3.3). The amount of DP 3 products after 24 h reaction was 100.1 g/L and it was comparable to the DP 3 fraction (95.2 g/L) obtained with *M. reukaufii* cell extract on 500 g/L sucrose [17]. By NMR analysis, several products with DP of 2 and 3 with α-1,1, α-1,3, α-1,4, and α-1,6 linkages such as maltulose, trehalulose, turanose, nigerose, isomaltose, isomaltulose, erlose, theanderose, melezitose, isomelezitose, and Glc-α-1,1-Fru-β-2,1-Glc were disclosed among the transglycosylation products of *Ba*AG2 (Table 1 and Table 2). Even though isomelezitose and trehalulose—the major transglycosylation products in *Metschikowia* sp. α-glucosidase reactions [16]—were among the dominant products, the ratios were significantly different from previously reported product profiles. Importantly, we also detected some novel α-1,1- and α-1,4-linked trisaccharide products characterized for the first time from an α-glucosidase transglycosylation reaction.

The enlarged space next to subsites +1 and +2 shown in Figure 6, panels b and c, can most likely accommodate either mono- or disaccharide acceptors for transglycosylation, and in different orientations. The NMR analysis of transglycosylation products indicates that fructose and glucose can both act as glycosyl acceptors, and they preferably bind at orientation enabling the formation of α-1,4 and α-1,1 linkages (Table 1). When sucrose is serving as an acceptor, its binding to the enzyme mostly results in synthesis of α-1,6 linkage (Table 2). Interestingly, analysis of sucrose transglycosylation products shows that it can be bound to +1 and +2 subsites of the *Ba*AG2 in two alternative orientations—either fructose or glucose moiety of sucrose binding the +1 subsite. Thus, our data suggest unusually relaxed (unspecific) binding at the acceptor site where only few specific contacts with the substrate are made.

## 5. Conclusions

*Blastobotrys* species represent a valuable biotechnological resource due to their thermo- and osmotolerance, oleaginous traits and enzyme secretion capacity. In addition, *B. adeninivorans* is able to efficiently utilize sucrose and several maltose-like substrates. We have shown the importance of the α-glucosidase *Ba*AG2 for metabolizing sucrose, maltose, malto-oligosaccharides, and even starch and glycogen. Importantly, *Ba*AG2 is able to synthesize a variety of rare di- and trisaccharides in considerable quantities. In this work, the product spectrum from sucrose transglycosylation reaction was studied in detail using NMR spectroscopy. A remarkable variety of glycosidic bonds, i.e., α-1,1, α-1,3, α-1,4, and α-1,6, were detected among transglycosylation products formed from sucrose by *Ba*AG2. Among those, the dominating products were candidate prebiotics maltulose, trehalulose, isomelezitose, erlose, and theanderose, which are also found in honey. Interestingly, a novel product 1-O^F^-glucosyl-sucrose (Glc-α-1,1-Fru-β-2,1-Glc) was also identified and assigned by NMR. The bioactive oligosaccharides hold an enormous potential in the food industry. Therefore, the biochemical and structural study of enzymes that are capable of producing these sugars from widely available and cheap substrates such as sucrose is certainly justified.

We solved here the two structures of the wild-type *Ba*AG2: one with acarbose and the other with glucose bound to the active site. Even though the properties of maltases from yeasts have been extensively studied over many decades, this is the first report on crystal structures of a yeast maltase. The overall structure and active site architecture of the enzyme was determined and the binding modes of both ligands were revealed. Expectedly, the *Ba*AG2 exhibited a catalytic domain with a (β/α)_8_-barrel (TIM-barrel) fold and two aspartates and a glutamate (Asp216, Glu274, Asp348) as catalytic triad residues. The active site cleft was found to be fairly wide, and next to the substrate-binding pocket at +1 and +2 subsites an enlarged space was detected that is absent from the isomaltase *Sc*IMA1. This space has several water molecules assisting substrate hydrolysis and it most likely accommodates mono- and disaccharides as acceptors for the transglycosylation reaction creating a variety of di- and trisaccharidic transglycosylation products. The involvement of substrate binding by a Glu (Glu309) in +2 and an Arg (Arg233) in +3 subsite was shown for the first time for α-glucosidases. Further studies on structure-function analysis of acceptor binding are needed to explain the formation of specific product spectrum of *Ba*AG2 and other yeast maltases but the structural data obtained in this work enables rational design of mutants with altered active-site pockets.

## Figures and Tables

**Figure 1 jof-07-00816-f001:**
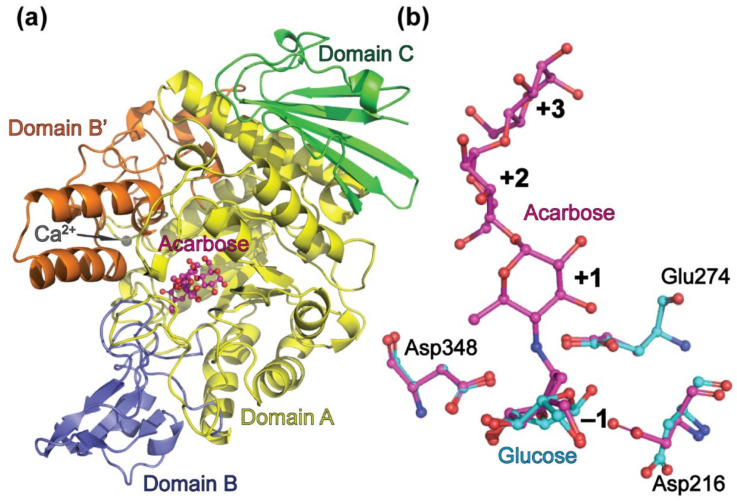
Overall structure of the *Ba*AG2. A cartoon of the *Ba*AG2-acarbose structure, with acarbose shown in the active center (**a**). Domain A is in yellow, domains B and B’ are in blue and orange, respectively, and domain C is in green. Calcium ion is shown in grey. Panel (**b**) shows amino acids of the catalytic triad (Asp216, Glu274, and Asp348) of *Ba*AG2 indicating different orientation of the side chain of Asp216 in the acarbose- and glucose-bound structures. The *Ba*AG2-acarbose structure is shown in magenta, the *Ba*AG2-glucose structure is in light blue. Structures were aligned and visualized with PyMol [39].

**Figure 2 jof-07-00816-f002:**
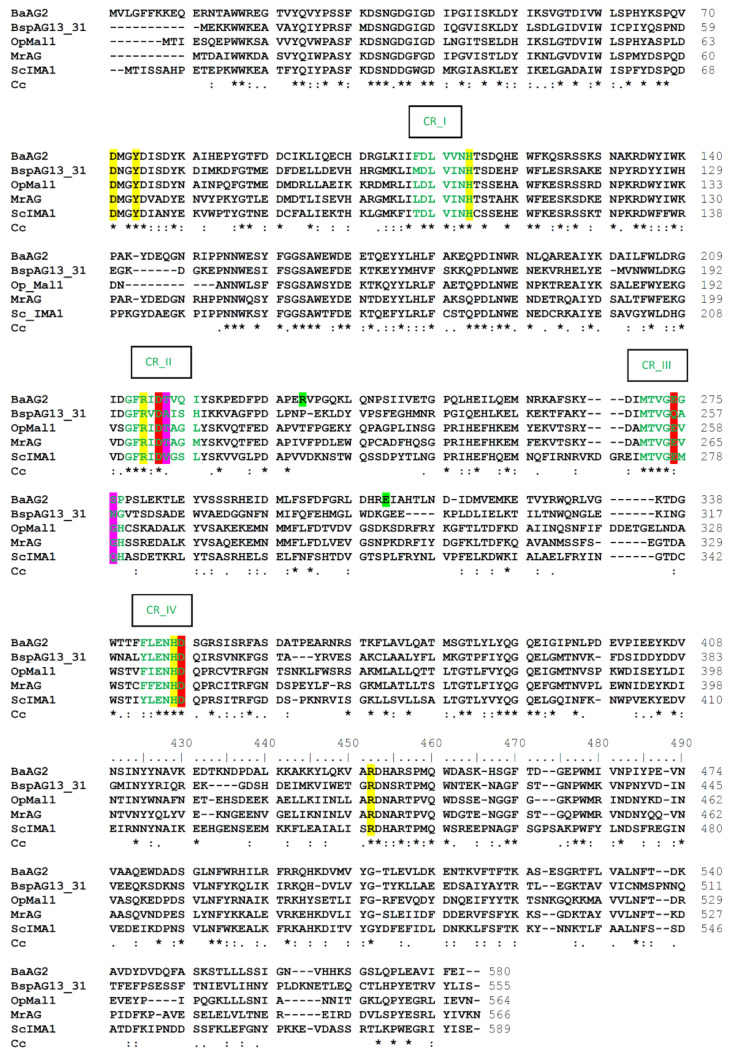
Clustal Omega alignment of amino acid sequences of *Ba*AG2 (GenBank: MZ467078, translation), BspAG13_31 (α-glucosidase of *Bacillus* sp. AHU2216; PDB: 5ZCE), *Op*Mal1 (maltase-isomaltase Mal1 of *O. polymorpha*; GenBank: XP_018213389.1, AWO14629.1), *Mr*AG (α-glucosidase of *M. reukaufii*; GenBank: QLP89119.1) and *Sc*IMA1 (isomaltase of *S. cerevisiae*; GenBank: NP_011803.3). Catalytic triad residues are shown with a red background, residues corresponding to *Ba*AG2 residues making contacts with the substrate at −1 subsite (see also Figure 4) with a yellow background, further binding sites residues of *Ba*AG2 with a green background, and residues equivalent to Gln279 and Val216 of *Sc*IMA1 determining substrate specificity have a magenta background. Four conserved regions (CR) of α-glucosidases as given by [49] on the basis of Taka-amylase are shown by green letters. Clustal consensus (Cc) is marked below the sequence indicating conservation — * positions with fully conserved residue; : positions with residues of strongly similar properties; . positions with residues of weakly similar properties.

**Figure 3 jof-07-00816-f003:**
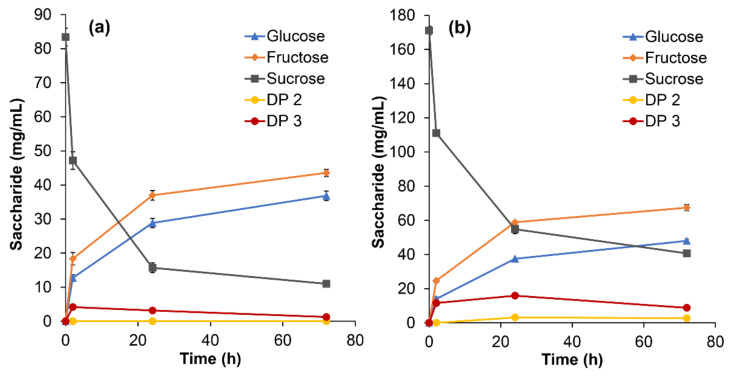
Transglycosylation of sucrose by *Ba*AG2. 20 µg/mL of the enzyme was reacted with 250 mM (**a**) or 500 mM (**b**) of sucrose. Samples from the reaction mixtures were withdrawn at designated time points, heated to terminate the reaction and analyzed for sugar composition by HPLC as described in Materials and Methods, Section 2.4 and Section 2.6. Concentrations of detected saccharides at each time point are shown. Products were specified using glucose, fructose, sucrose, isomaltose, and melezitose as references. Transglycosylation products with DP 2 (other than sucrose and its isomers) and DP 3 are marked with yellow or red circles, respectively. SDs of two to three HPLC measurements at each time point are indicated by error bars.

**Figure 4 jof-07-00816-f004:**
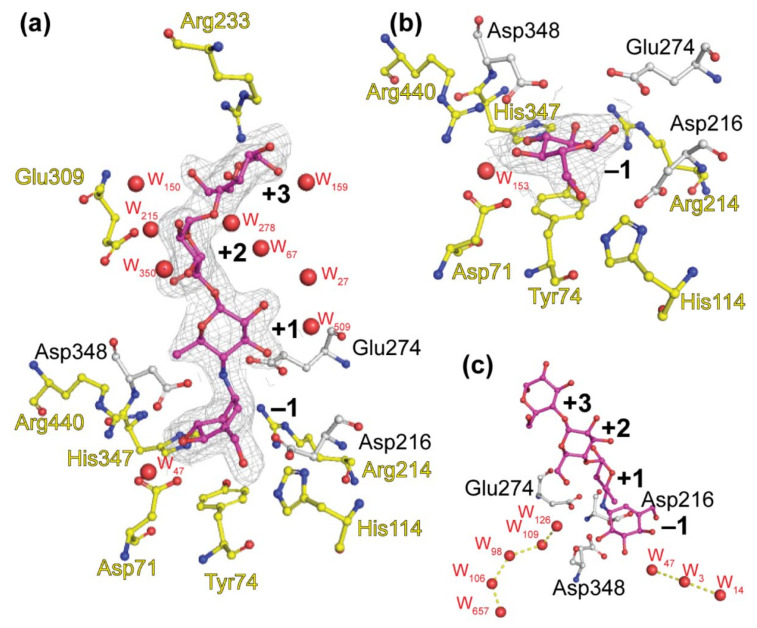
Binding of α-acarbose (**a**) and D-glucose (**b**) to the active center of the maltase *Ba*AG2. Water chains reaching the active center on *Ba*AG2-acarbose structure are shown on panel (**c**). Catalytic triad (Asp216, Glu274, and Asp348) residues are in grey, amino acids contributing to ligand binding are shown in yellow, and water molecules (W) are shown as red spheres. Acarbose and glucose are colored in magenta and final 2Fo-Fc electron density contoured at 1 σ. Binding subsites of acarbose are designated with bold numbers. Structures were visualized with PyMol [39].

**Figure 5 jof-07-00816-f005:**
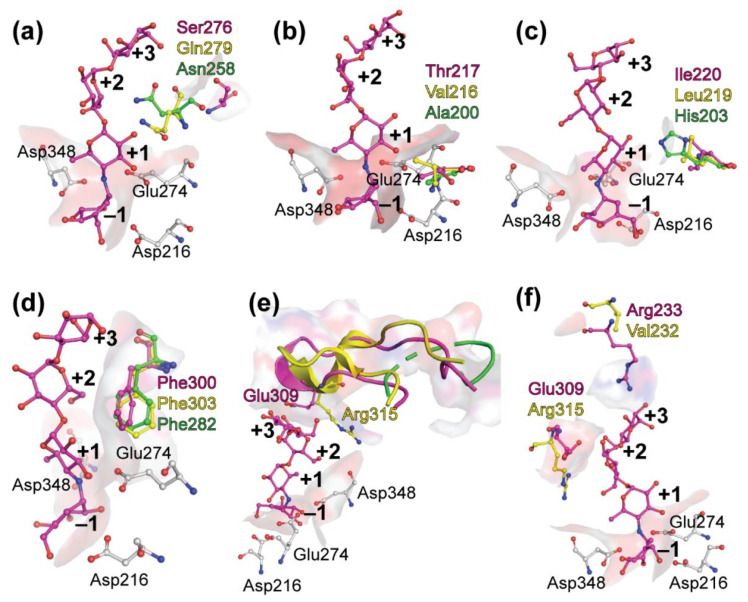
Amino acids participating in substrate binding among GH13 α-glucosidases. Positions of Ser276, Thr217, Ile220, Phe330, Glu309 and Arg233 of *Ba*AG2 from *B. adeninivorans* (in magenta) shown on panels **a–f,** respectively, were superimposed with the Glu277Ala mutant of *S. cerevisiae* isomaltase IMA1 (in yellow; PDB: 3AXH) and the Glu256Gln mutant of *Bacillus* sp. AHU2216 maltase BspAG13_31 (in green, PDB: 5ZCE). The catalytic triad (Asp216, Glu274 and Asp348) of *Ba*AG2 is shown by sticks in grey, and acarbose from the *Ba*AG2 structure is colored in magenta. The surface mode is applied to amino acids from *Ba*AG2-acarbose structure and subsites of catalytic cavity, from −1 to +3, are designated with bold numbers. Structures were aligned and visualized with PyMol [39].

**Figure 6 jof-07-00816-f006:**
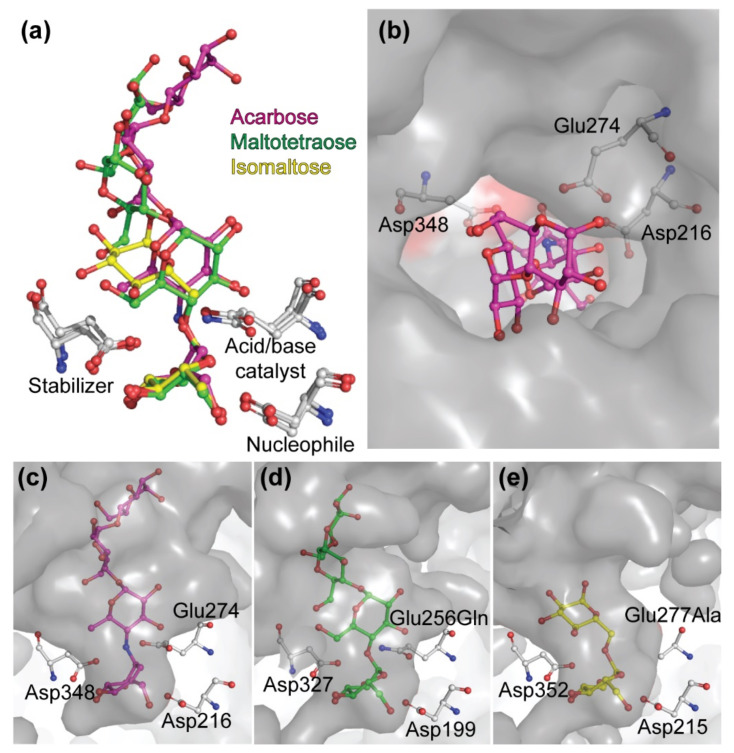
Substrate binding pocket of selected GH13 enzymes. Superimposed conformations of acarbose, maltotetraose and isomaltose (**a**). Top view of substrate binding cavity of the maltase *Ba*AG2 of *B. adeninivorans* with bound acarbose (magenta) (**b**). Side views of substrate binding modes of acarbose (panel **c**, magenta) to *Ba*AG2, maltotetraose (panel **d**, green) to BspAG13_31 mutant Glu256Gln of *Bacillus sp*. AHU2216 (PDB: 5ZCE) and isomaltose (panel **e**, yellow) to IMA1 mutant Glu277Ala of *S. cerevisiae* (PDB: 3AXH). Catalytic triad amino acids are marked by grey sticks. Structures were aligned and visualized with PyMol [39].

**Figure 7 jof-07-00816-f007:**
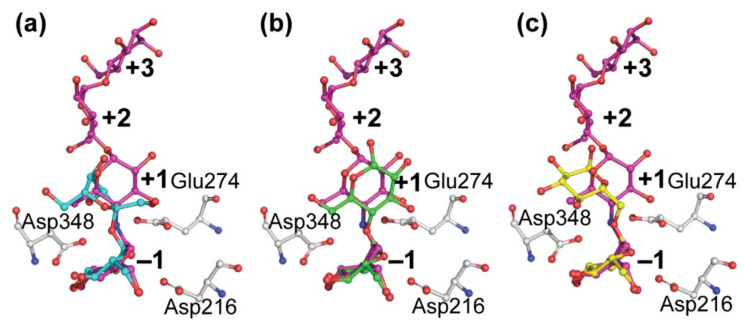
Positioning of acarbose (magenta) from *Ba*AG2 of *B. adeninivorans* superimposed with that of sucrose, maltose, and isomaltose. Sucrose (blue, **a**) originates from the structure of sucrose hydrolase from *B. mori* (PDB: 6LGF). Maltose (green, **b**) was taken from the structure of maltase from *Bacillus* sp. AHU2216 (PDB: 5ZCC) and isomaltose (yellow, **c**) is from the structure of isomaltase IMA1 of *S. cerevisiae* (PDB: 3AXH). The catalytic triad of *Ba*AG2 is in grey. Binding subsites of acarbose are labelled with bold numbers. Structures were aligned and visualized with PyMol [39].

**Table 1 jof-07-00816-t001:** Disaccharide proportions in transglycosylation reactions with 500 mM or 1500 mM sucrose by the *Ba*AG2. Sucrose isomers are shown in bold.

Saccharide	Composition and Linkage Type	500 mM Sucrose	1500 mM Sucrose
Sucrose—Substrate	Glc-α1,2-Fru	43.2%	69.4%
Maltulose	**Glc-α1,4-Fru**	23.9%	13.5%
Trehalulose	**Glc-α1,1-Fru**	16.4%	7.8%
Turanose	**Glc-α1,3-Fru**	5.9%	4.2%
Nigerose	Glc-α1,3-Glc	3.7%	1.9%
Maltose	Glc-α1,4-Glc	3.0%	1.0%
Isomaltose	Glc-α1,6-Glc	2.1%	1.7%
Isomaltulose (Palatinose)	**Glc-α1,6-Fru**	1.8%	0.4%

**Table 2 jof-07-00816-t002:** Trisaccharide proportions produced from 500 mM or 1500 mM sucrose by the *Ba*AG2. The sugar in each trisaccharide in bold was used for integration. The sucrose moiety in trisaccharides is underlined.

Saccharide	Composition and Linkage Type	500 mM Sucrose	1500 mM Sucrose
Isomelezitose *	**Glc-α1**,6-Fru-β2,1-Glc	32.2%	25.0%
Erlose	**Glc-α1**,4-Glc-α1,2-Fru	25.8%	31.7%
Theanderose *	**Glc-α1**,6-Glc-α1,2-Fru	22.7%	20.5%
Melezitose	**Glc-α1**,3-Fru-β2,1-Glc	9.6%	15.9%
1-O^F^-Glucosyl-Sucrose	**Glc-α1**,1-Fru-β2,1-Glc	3.7%	2.7%
Esculose	**Glc-α1**,3-Glc-α1,2-Fru	2.6%	0.8%
4-O^G^-Glucosyl-Turanose	**Glc-α1**,4-Glc-α1,3-Fru	2.0%	1.7%
4-O^F^-Glucosyl-Sucrose	**Glc-α1**,4-Fru-β2,1-Glc	1.4%	1.7%

* As the anomeric signals overlapped, the relative ratio of the two were determined using the signal arising from the 6-position of the 6-substituted Glc in theanderose and the 6-position of the 2,6-substituted Fru in isomelezitose.

## Data Availability

The data created during the study is presented in the article and in Appendix A. *Ba*AG2-encoding maltase gene sequence of *B. adeninivorans* CBS 8244 is available in GenBank under the accession no. MZ467078. The protein structure datasets created during the study are publicly available in Protein Data Bank: 7P01, 7P07.

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
