# Peer review of "Structural Insight into a Yeast Maltase—The BaAG2 from Blastobotrys adeninivorans with Transglycosylating Activity"

_jof, 2021, doi:10.3390/jof7100816_

Round 1

Reviewer 1 Report

In this manuscript Ernits et al. characterized the maltase protein BaAG2, belonging to the GH13 family, from the non-conventional yeast species B. adeninivorans. Specifically, the authors studied the transglycosylation activity of sucrose by BaAG2 and determined the mechanism of saccharide binding through crystallography experiments. The structural data shows the presence of a catalytic triad in the active site of BaAG2 which is similar to other maltases of GH13 family. However, the unique structural differences in the active center of BaAG2 allow it to hydrolyze and transglycosylate a variety of sugars. This feature is important because the authors show that transglycosylation of sucrose by BaAG2 results in production of numerous di- and tri-saccharides, which are considered probiotics and has enormous potential in the food industry. This further increases the impact of the mechanistic details obtained from the structural experiments and open new avenues of research. The experiments in the manuscript are really well described and the results and discussion show the in depth analysis that was performed by the authors. This is an excellent and well written manuscript, but due to the nature of the study the manuscript will not appeal to the general audience, which is not a critique of the manuscript. Overall, this is a very well done study and I recommend that the manuscript be accepted in its current form.

Author Response

Response to Reviewer 1 Comments

In this manuscript Ernits et al. characterized the maltase protein BaAG2, belonging to the GH13 family, from the non-conventional yeast species B. adeninivorans. Specifically, the authors studied the transglycosylation activity of sucrose by BaAG2 and determined the mechanism of saccharide binding through crystallography experiments. The structural data shows the presence of a catalytic triad in the active site of BaAG2 which is similar to other maltases of GH13 family. However, the unique structural differences in the active center of BaAG2 allow it to hydrolyze and transglycosylate a variety of sugars. This feature is important because the authors show that transglycosylation of sucrose by BaAG2 results in production of numerous di- and tri-saccharides, which are considered probiotics and has enormous potential in the food industry. This further increases the impact of the mechanistic details obtained from the structural experiments and open new avenues of research. The experiments in the manuscript are really well described and the results and discussion show the in depth analysis that was performed by the authors. This is an excellent and well written manuscript, but due to the nature of the study the manuscript will not appeal to the general audience, which is not a critique of the manuscript. Overall, this is a very well done study and I recommend that the manuscript be accepted in its current form.

Response: We kindly thank the Reviewer 1 for thorough work and a very positive feedback to the manuscript. We very much appreciate the fair insight to the manuscript and pointing out the potential limitations in reaching wide audiences due to the specificity of the study.

According to the comments made by Reviewer 2 the abstract of the manuscript was partially rewritten to enhance clarity and set the focus more on obtained results. In the manuscript text some sections were rephrased to enhance clarity in the regard of structure-function relationships among maltases, isomaltases and maltase-isomaltases of different origin. Also, reasons for the use of NMR methodology were highlighted. The text has been carefully checked for inconsistencies and errors in grammar and spelling. The changes made in the manuscript text can be followed by “Track changes” in the Word file.

Reviewer 2 Report

The work entitled “Structural Insight into a Yeast Maltase – The BaAG2 from Blastobotrys adeninivorans with Transglycosylating Activity”by Ernits et al., has been reviewed. The work described the in the current study, transglycosylation of sucrose was studied in detail. The chemical entities of sucrose-derived oligosaccharides were determined using nuclear magnetic resonance. Allthoruh the work potentially interesting to the reader that need a major revision before it considered for the publication in the Journal of Fungi.  

transglycosylation of sucrose

Cloning of maltase gene from type strain of B. adeninivorans CBS 8244 and producing BaAG2 protein by heterologous expression

Comparison of BaAG2 amino acid sequence with two homologues of strains LS3 and TMCC 70007

BaAG2 had the highest identity (97.59%) to putative maltase of TMCC 70007 and slightly lower identity (95.86%) to LS3. The predicted identity between the two maltases in strains TMCC 70007 and LS3 is 95.96%

In general, the overall structure of BaAG2 is very similar to that of bacterial maltase, S. cerevisiae isomaltase, bacterial dextran glucosidase and α-amylase of different origins - it has four domains.

Using maltose as a substrate, BaAG2 exhibits significant glycosylation activity to synthesize trisaccharides-maltotriose and phanose, and α-1,6-linked disaccharide isomaltose.

HPLC analysis is used to quantify sugars in the course of the reaction.

Attempted in-depth structural analysis of transglycosylation products using NMR.

  1. The abstract is deals the background about 6 lines and additionally two lines to describe the objective of the study. That leads to loss in presenting the current results in abstract. Therefore, I suggest the author to rephrase the TOC of abstract in attractive manner and better understanding of present work without dilution.
  2. In the introduction the line 56-57, the author need check the sentence “As a result, rare oligo-saccharides also found in honey [18–20], e.g. erlose, melezitose, theanderose, isomaltose, maltulose and turanose, can be produced”.
  3. In the case of sequences showing differences between maltase enzymes, what role of amino acids differs and what effect does this have on enzyme activity?
  4. Reasons for using NMR for in-depth structural analysis of transglycosylation products
  5. The structure of BaAG2 is similar to that of bacterial maltase, S. cerevisiae isomaltase, bacterial dextran glucosidase, and α-amylase of different origins, and what is the relationship between structural similarity and activity similarity?

Author Response

Response to Reviewer 2 Comments

The work entitled “Structural Insight into a Yeast Maltase – The BaAG2 from Blastobotrys adeninivorans with Transglycosylating Activity”by Ernits et al., has been reviewed. The work described the in the current study, transglycosylation of sucrose was studied in detail. The chemical entities of sucrose-derived oligosaccharides were determined using nuclear magnetic resonance. Allthoruh the work potentially interesting to the reader that need a major revision before it considered for the publication in the Journal of Fungi.

Response: We kindly thank the Reviewer 2 for thorough work and suggestions for improvement of the manuscript. Please find our point-by-point responses below to the specific comments raised by the Reviewer 2. The abstract as well as some parts of the rest of the manuscript have been amended according to the remarks and recommendations. The changes made in the manuscript text can be followed by “Track changes” in the Word file.   

transglycosylation of sucrose

Cloning of maltase gene from type strain of B. adeninivorans CBS 8244 and producing BaAG2 protein by heterologous expression

Comparison of BaAG2 amino acid sequence with two homologues of strains LS3 and TMCC 70007

BaAG2 had the highest identity (97.59%) to putative maltase of TMCC 70007 and slightly lower identity (95.86%) to LS3. The predicted identity between the two maltases in strains TMCC 70007 and LS3 is 95.96%

In general, the overall structure of BaAG2 is very similar to that of bacterial maltase, S. cerevisiae isomaltase, bacterial dextran glucosidase and α-amylase of different origins - it has four domains.

Using maltose as a substrate, BaAG2 exhibits significant glycosylation activity to synthesize trisaccharides-maltotriose and phanose, and α-1,6-linked disaccharide isomaltose.

HPLC analysis is used to quantify sugars in the course of the reaction.

Attempted in-depth structural analysis of transglycosylation products using NMR.

  1. The abstract is deals the background about 6 lines and additionally two lines to describe the objective of the study. That leads to loss in presenting the current results in abstract. Therefore, I suggest the author to rephrase the TOC of abstract in attractive manner and better understanding of present work without dilution.

Response 1: We thank the Reviewer for pointing this out and we completely agree with the comment. The abstract of the manuscript was partially rewritten to enhance its overall clarity and to set the focus on obtained results in the relevant context.

  1. In the introduction the line 56-57, the author need check the sentence “As a result, rare oligo-saccharides also found in honey [18–20], e.g. erlose, melezitose, theanderose, isomaltose, maltulose and turanose, can be produced”.

Response 2: We thank the Reviewer for the remark. It has been shown by application of various chromatography and spectroscopy methods including nuclear magnetic resonance, gas-chromatography mass-spectrometry and HPLC that honey samples from different parts of the world contain oligosaccharides such as erlose, melezitose, theanderose, isomaltose, maltulose and turanose (please see the references 18-20 in the main text). These above-mentioned oligosaccharides have also been identified as transglycosylation products of some fungal α-glucosidases (please see the references 1,14-17, 21 in the main text). But it has to be noted that there are only a few publications available describing in detail the sucrose-derived transglycosylation products of maltases. The sentences (lines 56-59 in the manuscript file without Track Changes) were rephrased for clarity. 

  1. In the case of sequences showing differences between maltase enzymes, what role of amino acids differs and what effect does this have on enzyme activity?

Response 3: We thank the reviewer for the relevant comment. The manuscript has been revised to discuss the determinants of different activities and substrate preferences among maltases, isomaltases and maltase-isomaltases in a more comprehensive way. In chapters 4.1 and 4.2 some sections addressing the structural determinants of different activities (between lines 523-527, 595-660) were rephrased accordingly to enhance clarity of the text.     

  1. Reasons for using NMR for in-depth structural analysis of transglycosylation products

Response 4: Nuclear magnetic resonance (NMR) is considered as one of the benchmark methods to analyse composition and linkage types of various saccharides. Chromatography and mass-spectrometry methods often fail to distinguish between isomers of the molecules. This was also the case in the current study as the applied HPLC method which is suitable for quantification of monosaccharides and short-chain oligosaccharides, was insufficient for detailed analysis of di- and trisaccharide mixture produced from sucrose as the isomers coeluted from the HPLC column. To identify the entities of trisaccharides a large set of standards is needed. In addition, without using the NMR method, we would not have been able to identify previously undescribed trisaccharides (please see Table 2). Based on our knowledge the current work is the most thorough analysis of a maltase transglycosylation products on sucrose where a variety of compounds was detected. The purposes of using NMR to confirm the structural identities of transglycosylation products were highlighted in the chapter 3.3 (lines 361-366) and the corresponding section was rephrased to enhance clarity.

  1. The structure of BaAG2 is similar to that of bacterial maltase, S. cerevisiae isomaltase, bacterial dextran glucosidase, and α-amylase of different origins, and what is the relationship between structural similarity and activity similarity?

Response 5: The aim of our study was to determine the structural features of a yeast maltase (BaAG2) and discuss the substrate specificity determinants of maltases, isomaltases and maltase-isomaltases of different origin (summarized in lines 85-89). In this work we are not concentrating on comparing maltases to the rest of α-glucosidases with various activities. Although, the overall structures of catalytic domains of all GH13 and GH13 enzymes are similar as they share a TIM-barrel fold and catalytic triad amino acids are fully conserved, the identity values between protein sequences are very low across the GH13 and GH31 enzymes and there are some severe differences in structures (including the number of domains, organisation and length of loops) making it very difficult to pinpoint the exact determinants. The proposed topic on detailed comparison of structural determinants and similarities/differences in activity of these enzymes would be a very interesting topic for a comprehensive review paper but we believe it is out of the scope of the current manuscript.

Round 2

Reviewer 2 Report

Accept